# LeMoLE: LLM-Enhanced Mixture of Linear Experts for Time Series Forecasting

## Abstract

Recent research has shown that large language models (LLMs) can be effectively used for real-world time series forecasting due to their strong natural language understanding capabilities. However, aligning time series into semantic spaces of LLMs comes with high computational costs and inference complexity, particularly for long-range time series generation. Building on recent advancements in using linear models for time series, this paper introduces an LLM-enhanced mixture of linear experts for precise and efficient time series forecasting. This approach involves developing a mixture of linear experts with multiple lookback lengths and a new multimodal fusion mechanism. The use of a mixture of linear experts is efficient due to its simplicity, while the multimodal fusion mechanism adaptively combines multiple linear experts based on the learned features of the text modality from pre-trained large language models. In experiments, we rethink the need to align time series to LLMs by existing time-series large language models and further discuss their efficiency and effectiveness in time series forecasting. Our experimental results show that the proposed LeMoLE model presents lower prediction errors and higher computational efficiency than existing LLM models.

## 1 Introduction

Long-term time series forecasting (LTSF) is a significant challenge in machine learning due to its wide range of applications. It has been important in various domains such as weather modeling (Ma et al., 2023; Lin et al., 2022), traffic flow management (Lv et al., 2014), and financial analysis (Abu-Mostafa & Atiya, 1996). Traditional statistical models like ARIMA (Box & Pierce, 1970) and exponential smoothing (Gardner Jr, 1985) have served as the foundation for forecasting tasks for decades. However, these models often struggle to handle the complexities arising from real-world applications, such as non-linearity, high dimensionality, and intricate temporal dynamics. In recent years, deep learning models have emerged as a breakthrough in forecasting, revolutionizing accuracy and efficiency. These models can remarkably capture complex temporal patterns and interactions within the data. By leveraging the power of deep learning, they excel in forecasting tasks by effectively learning from large-scale datasets.

It is intriguing to note that while deep models (e.g., transformer-based models) have gained popularity and achieved significant success in various fields like computer vision, natural language processing, and time series research, they usually come at the cost of extensive computational burdens. Recent empirical studies have revealed scenarios where simpler and more computationally efficient linear-based models outperform complex deep learning models. Models like DLinear (Zeng et al., 2023) and RLinear (Li et al., 2023) have demonstrated superior performance. Linear models have proven effective in time series forecasting due to their capacity to capture and leverage the linear relationships inherent in many time series datasets. By exploiting these linear relationships, linear-based models can provide competitive predictions while maintaining computational efficiency. While linear-based models have demonstrated strengths in certain time series forecasting scenarios, it is important to acknowledge their limitations:

i) *Non-linear patterns*: Real-world time series data often exhibit non-linear patterns resulting from complex underlying mechanisms, such as variable interactions or abrupt regime shifts. Linear models may struggle to capture and model these non-linear relationships effectively (Chen et al., 2023; Ni et al., 2024; Lin et al., 2024).

ii) *Long-range dependencies*: Linear models might face difficulties handling long-term dependencies within time series data. As the dependency structure becomes more intricate and extends over longer periods, the effectiveness of linear models diminishes (Nie et al., 2023a; Liu et al., 2024c).

Therefore, the challenge of *developing a powerful prediction model that retains the high efficiency of linear models* remains an open question.

A mixture of linear experts is a promising solution to build such a model. Intuitively, multiple linear experts can convert the original nonlinear time series prediction into several component prediction problems. For example, some experts focus on trends, while others handle seasonals, or some deal with short-term patterns while others learn long-term patterns. For example in (Ni et al., 2024), Mixture-of-Linear-Experts (MoLE) is proposed to train multiple linear-centric models (i.e., experts) to *collaboratively* predict the time series. Additionally, a router model, which accepts a timestamp embedding of the input sequence as input, learns to weigh these experts adaptively. This allows that different experts specialize in different periods of the time series.

In addition, incorporating multimodal knowledge into predictive models is also a promising solution. Recently, there has been a significant surge of interest in multimodal time series forecasting. For example, TimeLLM (Jin et al., 2024) aims to align the modalities of time series data and natural language such that the capabilities of pretrained large language model (LLM) from natural language process (NLP) can be activated to model time series dynamics. In practice, the alignment of multimodality in time series forecasting can be easily achieved by fine-tuning the input and output layers. In this way, both time series and non-time series data (such as text data)

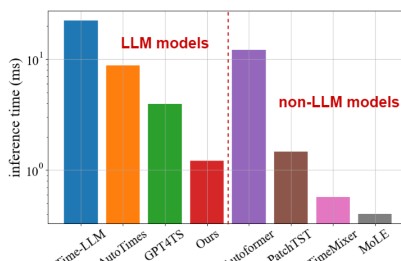

Figure 1: Inference time on `ETTh1`.

can be jointly inputted to LLM for multimodal time series forecasting. Although such alignment-based LLMs have shown improvement in time series forecasting tasks, compared to linear models, they are not very effective and suffer from slow inference speed (Liu et al., 2024b) as they have to use large language model as time series predictor. Figure 1 shows inference efficiency comparisons.

Motivated by the above-related works, in this paper, we propose LeMoLE for Time Series Forecasting. LeMoLE refers to an LLM-enhanced mixture of linear experts. Different from the Mixture-of-Linear-Experts (MoLE) (Ni et al., 2024), the proposed LeMoLE enhances ensemble diversity by leveraging multiple linear experts with varying lookback window lengths. This strategy is simple yet effective. Intuitively, this improvement encourages the experts to effectively handle both short-term and long-term temporal patterns in historical data. Moreover, LeMoLE incorporates informative multimodal knowledge from global and local text data during the ensemble process of the multiple linear experts. This adaptive approach allows LeMoLE to allocate specific experts for specific temporal patterns, enhancing its flexibility and performance. We introduce a pre-trained large language model for extracting text representations to improve the fusion of outputs from multiple linear experts and text knowledge. Additionally, to incorporate static and dynamic text information, we incorporate two conditioning modules based on the well-known FiLM (Feature-wise linear modulation) conditioning layer (Perez et al., 2018). Consequently, the proposed LLM-enhanced mixture of linear experts enables more flexible and effective long-range predictions than alignment-based time series LLM models.

The main contributions of our work are summarized as follows:

i) We present an LLM-enhanced mixture of linear experts called LeMoLE. To the best of our knowledge, it is the first work on improving linear time series models based on mixture-of-expert learning and multimodal learning.

ii) We introduce linear experts with varying lookback window lengths to enhance ensemble diversity and incorporate two novel conditioning modules based on FiLM (Feature-wise Linear Modulation) to effectively integrate global and local text data adaptively.

iii) We rethink existing large language models for time series and compared several recent state-of-the-art prediction networks on long-term forecasting and few-shot tasks. The results demonstrate the effectiveness of the proposed LeMoLE in terms of accuracy and efficiency.

## 2 RELATED WORK

### 2.1 LINEAR MODELS AND LINEAR ENSEMBLE MODELS

While transformer-based models (Zhou et al., 2022a; Nie et al., 2023a; Wu et al., 2021) have been successful in Long-Term Time Series Forecasting (LTSF), (Zeng et al., 2023) questioned their universal superiority and suggested simpler architectural approaches like DLinear and NLinear. DLinear (Zeng et al., 2023) decomposes time series into trend and season branches and uses linear models for forecasting. Subsequent research by (Li et al., 2023) further confirmed the potential of linear-centric models like RLinear and RMLP, which outperformed PatchTST (Nie et al., 2023a) in specific benchmarks. Based on linear-based models and research focusing on the frequency domain, FITS (Xu et al., 2024) operates within the complex frequency domain. Although linear models are efficient, they are still limited in high-nonlinear time series (Chen et al., 2023; Ni et al., 2024). Related ensemble linear models, such as TimeMixer (Wang et al., 2024) mixing the decomposed season and trend components of time series from multiple resolutions. Then, multiple predictors are utilized to project the resolution features for the final prediction. Based on a mixture of experts, MoLE (Ni et al., 2024) applies multiple linear experts for forecasting, which is based on a router module to adaptively reweigh experts' outputs for the final generation. The proposed LeMoLE is different from them due to its multimodal fusion mechanism.

### 2.2 LLM-BASED MULTIMODAL FORECASTING

Pre-trained foundation models, such as large language models (LLMs), have driven rapid progress in natural language processing (NLP) (Radford et al., 2019; Brown, 2020; Touvron et al., 2023) and multimodal modeling (Caffagni et al., 2024; Hu et al., 2024). Several works have tried to transfer LLMs' capabilities of other modalities to advance time series forecasting. However, the main challenges lie in discussing the relationships between the two modalities, time series and text. Some previous works claim that aligning them is important and useful for multimodal forecasting. LLM4TS (Chang et al., 2023) use a two-stage fine-tuning process on the LLM, first supervised pre-training on time series, then task-specific fine-tuning. Zhou et al. (2024) leverages pre-trained language models without altering the self-attention and feedforward layers of the residual blocks. It is fine-tuned and evaluated on various time series analysis tasks to transfer knowledge from natural language pre-training. Jin et al. (2024) reprograms the input time series with text prototypes before feeding it into the frozen LLM to align the two modalities. Conversely, AutoTimes (Liu et al., 2024b) states the aligning is overlooked, resulting in insufficient utilization of the LLM potentials. It presents token-wise prompting that utilizes corresponding timestamps and then concatenates the time and prompt features as the multimodal input.

Although these LLM-based time series methods have improved, their main limitation is their efficiency compared with lightweight models like linear-based models. In this work, we rethinnk the use of large language models for time series and strive to develop a more efficient and effective LLM-enhanced prediction model.

## 3 LeMoLE: LLM-ENHANCED MIXTURE OF LINEAR EXPERTS

**Problem formulation.** Given a lookback window $\mathbf{X}_{1:T} \in \mathbb{R}^{T \times C}$ ($T$ is the length of history observations and $C$ is the number of variables), a task of time series forecasting aims to train a model $\mathcal{F}$ to predict its future values in a forecast window $\mathbf{X}_{T+1:T+H}$. Ideally, an optimal model $\mathcal{F}^*$ builds a (nonlinear) mapping between the lookback window and the forecast window:

$$\mathbf{X}_{T+1:T+H} = \mathcal{F}^*(\mathbf{X}_{1:T}). \tag{1}$$

However, the underlying temporal dynamics tend to be highly complex in terms of real-world time series characteristics. Consequently, training $\mathcal{F}$ to approximate $\mathcal{F}^*$ solely based on the lookback window becomes exceedingly challenging. Incorporating multimodal knowledge (such as time series-related text data) is a promising solution (Jin et al., 2024) to help time series forecasting. This work considers the text-enhanced time series forecasting scenes, where a static prompt (denoted as $\mathbf{P}_S$) and a dynamic prompt (denoted as $\mathbf{P}_D$) are processed by a pretrained large language model, and the extracted text features are used to enhance the time series prediction model. Formally,

$$\hat{\mathbf{X}}_{T+1:T+H} = \mathcal{F}(\mathbf{X}_{1:T}, \mathbf{P}_D, \mathbf{P}_S). \tag{2}$$

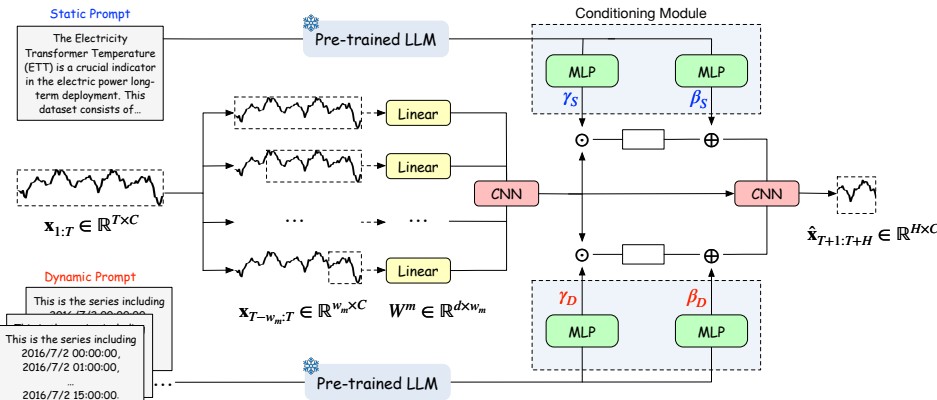

Figure 2: The proposed LeMoLE is based on a mixture of linear experts with different look-back lengths. We effectively incorporate (static and dynamic) multimodal knowledge into our approach by leveraging two frozen large language models (LLMs). The conditioning module associated with each LLM plays a crucial role in activating and enhancing our multi-expert prediction network. Finally, a lightweight CNN produces future predictions.

Here, $\hat{\mathbf{X}}$ denotes the estimation output of the forecast window. Figure 2 illustrates the proposed LeMoLE. Note that rather than simply combining multiple linear experts with the same lookback lengths as in (Ni et al., 2024), we set different lookback lengths for our linear experts. This allows different experts to focus on various short-term and long-term temporal patterns. This section will formally elaborate on each component in the proposed model.

## 3.1 MIXTURE OF LINEAR EXPERTS

Linear models have demonstrated effectiveness in time series forecasting (Zeng et al., 2023). However, due to their inherent simplicity, they are still limited to complex non-periodic changes in time series patterns (Ni et al., 2024). In the proposed LeMoLE, we introduce a mixture of linear experts with different lookback lengths to model both short-term and long-term temporal patterns.

Mathematically, let the number of experts be $M$. Given a time series window $\mathbf{X}_{1:T}$, we generate its $M$ views for $M$ experts respectively. For the $m$th expert ($m = 1, 2, \ldots, M$), we have the input as $\mathbf{X}_{T-w_m:T}$. Here, $w_m$ is the window length for the $m$th expert (we assume $w_1 \geq w_2 \geq \cdots \geq w_M$). Then we can obtain the prediction of the $m$th expert by

$$\mathbf{Y}^{(m)} = \mathbf{W}_m \mathbf{X}_{T-w_m:T} + \mathbf{b}_i, \tag{3}$$

where $m = 1, \ldots, M$, $\mathbf{W}_m \in \mathbb{R}^{H \times w_m}$ and $\mathbf{b}_m \in \mathbb{R}^{H \times C}$ are trainable expert-specific parameters. Based on Equation (??), we can obtain $M$ prediction output from $M$ linear experts, denoted by $\{Y^{(1)}, Y^{(2)}, \ldots, Y^{(M)}\}$. All of these outputs are with the same sizes of $H \times C$.

## 3.2 LLM-ENHANCED CONDITIONING MODULE

Prompting serves as a straightforward yet effective approach to task-specific activation of LLMs. To leverage abundant multimodal knowledge to help time series forecasting, it is essential to design appropriate text prompts and the corresponding conditioning module to activate our multi-expert prediction network.

In time series data, there are two important types of text information that describe temporal dynamics. The first type is static text, which typically provides global information about the time series dataset, such as data source descriptions. The second type of text is dynamic and time-dependent, including information like time stamps, weather conditions, or other external environmental factors. To incorporate these two types of text data into the prediction network, we create static and dynamic prompts and use a pretrained language model to obtain their corresponding representations.

**Static prompt.** Figure 5 (left) in Appendix B shows a static prompt example we used on the `ETTh` dataset. It is about the data source description. Specifically, it includes what, where, and how the

data was collected. Also, it contains the meanings of variables in the multivariate time series. This information helps understand and assess the reliability and relevance of the particular prediction tasks. We assume the static prompt $\mathbf{P}_S$ contains the $L_S$ length of texts (including punctuation marks). To facilitate the LLM ability of text understanding, the LLM encoder denoted as $\mathcal{LLM}(\cdot)$ is utilized to obtain the text representation vector $\mathbf{Z}_S \in \mathbb{R}^{L_S \times d_{llm}}$, i.e.

$$\mathbf{Z}_S = \mathcal{LLM}(\mathbf{P}_S). \tag{4}$$

where $d_{llm}$ is the dimension of the LLM encoder $\mathcal{LLM}$ token embeddings.

**Dynamic prompt.** Distinct from the static prompt, the timestamps in the datasets indicate when the observations were recorded. We follow AutoTimes (Liu et al., 2024b) to use the timestamps as related dynamic text data and design our dynamic prompt as in Figure 5 (right) in Appendix B. We aggregate textual covariates $\mathbf{T}_{T-w_1}, \ldots, \mathbf{T}_T$ to generate the dynamic prompt as $\mathbf{P}_D \in \mathbb{R}^{L_D \times 1}$. Formally, it is given by $\mathbf{P}_D = Prompt([\mathbf{T}_{T-w_1}, \mathbf{T}_{T-w_1+1}, \ldots, \mathbf{T}_T])$, where $w_1$ is the maximum lookback length in all experts. Then by LLM, the dynamic prompt is encoded into representations $\mathbf{Z}_D \in \mathbb{R}^{L_D \times d_{llm}}$ by

$$\mathbf{Z}_D = \mathcal{LLM}(\mathbf{P}_D). \tag{5}$$

## 3.3 Conditioning Module

After obtaining the representations $\mathbf{Z}_S \in \mathbb{R}^{L_S \times d_{llm}}$ and $\mathbf{Z}_D \in \mathbb{R}^{L_D \times d_{llm}}$ from the static prompt and dynamic prompt respectively, we can use them as conditions to activate our multi-expert prediction network. Specifically, we first introduce two conditioning modules to fuse $\mathbf{Z}_S$ and $\mathbf{Z}_D$ respectively and then use light-weight CNN blocks to summarize all branches to get the final prediction.

The proposed conditioning module is based on the popular conditioning layer, FiLM (Perez et al., 2018). First, we use a CNN to map the multi-linear experts' outputs $\{\mathbf{Y}^{(1)}, \mathbf{Y}^{(2)}, \ldots, \mathbf{Y}^{(M)}\}$ into a tensor $\mathbf{Y}$ of $H \times C$, say $\mathbf{Y} = \text{CNN}([\mathbf{Y}^{(1)}; \mathbf{Y}^{(2)}; \ldots; \mathbf{Y}^{(M)}])$. Then, we fuse the static representation $\mathbf{Z}_S \in \mathbb{R}^{L_S \times d_{llm}}$ with $\mathbf{Y}$ by

$$\mathbf{Y}'_S = \gamma_S \odot \mathbf{Y} + \beta_S, \tag{6}$$

where $\gamma_S = \text{Linear}^t_{S,1} \circ \text{Linear}^c_{S,1}(\mathbf{Z}_S)$, $\beta_S = \text{Linear}^t_{S,2} \circ \text{Linear}^c_{S,2}(\mathbf{Z}_S)$. Here, $\text{Linear}^t$ is the linear mapping to change the time dimension from $L_S$ to $H$. $\text{Linear}^c$ changes the channel dimension from $d_{llm}$ to $C$. Finally, we have $\gamma_S \in \mathbb{R}^{H \times C}$, $\beta_S \in \mathbb{R}^{H \times C}$, and the fused output $\mathbf{Y}'_S \in \mathbb{R}^{H \times C}$.

Similarly, when using dynamic representation $\mathbf{Z}_D$ as condition, we have

$$\mathbf{Y}'_D = \gamma_D \odot \mathbf{Y} + \beta_D, \tag{7}$$

where $\gamma_D = \text{Linear}^t_{D,1} \circ \text{Linear}^c_{D,1}(\mathbf{Z}_D)$, $\beta_D = \text{Linear}^t_{D,2} \circ \text{Linear}^c_{D,2}(\mathbf{Z}_D)$. Here, we obtain output $\mathbf{Y}'_D \in \mathbb{R}^{H \times C}$. Finally, we get the final prediction $\hat{\mathbf{Y}}$ by

$$\hat{\mathbf{Y}} = \text{CNN}^{\text{final}}([\mathbf{Y}; \mathbf{Y}'_S; \mathbf{Y}'_D]). \tag{8}$$

Given the final prediction $\hat{\mathbf{Y}}$, we can minimize the distance (e.g., mean square errors) between the ground truths $\mathbf{X}_{T+1:T+H}$ and predictions $\hat{\mathbf{Y}}$ to train the whole network in an end-to-end way

$$\mathcal{L} = ||\mathbf{x}_{T+1:T+H} - \hat{\mathbf{Y}}||^2_2. \tag{9}$$

The pseudocode for the training procedures of the backward denoising process can be found in Appendix A.

**Extension to frequency domain.** In the proposed LeMoLE, we introduce linear experts with varying lookback window lengths to enhance ensemble diversity. In this section, drawing inspiration from a recent frequency-based linear model known as FITS (Xu et al., 2024) (Frequency Interpolation Time Series Analysis Baseline), we propose an extension of LeMoLE called LeMoLE-F, where each linear expert is implemented using FITS. Consequently, we can rename the original LeMoLE in the time domain as LeMoLE-T. The setup of lookback window lengths of LeMoLE-F is the same as that in LeMoLE-T. In LeMoLE-F, each linear expert takes the input as a frequency domain projection of a specific lookback window. This projection is achieved by applying a real FFT (Fast Fourier Transform). Subsequently, a single complex-valued linear layer is used to interpolate the frequencies. To revert the interpolated frequency back to the time domain and obtain the output of the linear expert, zero padding and an inverse real FFT are applied.

## 4 EXPERIMENT

To verify the proposed LeMoLE model's effectiveness and efficiency, we conducted extensive experiments to dicsuss the following research questions. In Appendix F, we further provided the visualization results about using the proposed LeMoLE on real-world time series.

**RQ1:** How deos LeMoLE perform on long-range prediction and few-shot learning scenarios?
**RQ2:** Is multimodal knowledge, specifically text features, always useful on various datasets?
**RQ3:** What about using linear experts in the frequency domain?
**RQ4:** What are the effects of the hyperparameter sensitivity?
**RQ5:** Is LeMoLE computationally efficient compared to existing LLM-based time series models?

### 4.1 EXPERIMENTAL SETTINGS

**Datasets.** We conider four commonly-used real-world datasets (Jin et al., 2024; Wu et al., 2023): `ETTh1`, `ETTm1`, `Electricity` (ECL), and `Traffic` datasets. As in (Liu et al., 2022b), we use the Augmented Dick-Fuller (ADF) test statistic (Elliott et al., 1996) to evaluate if they are non-stationary. The null hypothesis is that the time series is not stationary (has some time-dependent structure) and

Table 1: Evaluation of non-stationarity by the Augmented Dick-Fuller (ADF) test. A higher ADF test statistic indicates a lower stationarity degree, meaning the distribution is less stable.

|  | Traffic | Electricity | ETTh1 | ETTm1 |
|---|---|---|---|---|
| ADF statistic | -2.801 | -2.797 | -2.571 | -1.734 |
| p-value | 0.005 | 0.006 | 0.099 | 0.414 |

can be represented by a unit root. The test statistic results are shown in Table 1. As can be seen, with a threshold of 5%, `ETTm1` and `ETTh1` are considered non-stationary. More details about datasets can be found in Appendix C.

**Baselines.** We compare our method with the recent strong time series models, including i) *CNN-based models*: FiLM (Zhou et al., 2022b), TimesNet (Wu et al., 2023); ii) *Linear models*: LightTS (Zhang et al., 2022), DLinear (Zeng et al., 2023), TSMixer (Chen et al., 2023), SparseTSF (Lin et al., 2024), TimeMixer (Wang et al., 2024), FITS (Xu et al., 2024) and MoLE (Ni et al., 2024); iii) *Transformers*: Informer (Zhou et al., 2021), Autoformer (Wu et al., 2021), PatchTST (Nie et al., 2023b), iTransformer (Liu et al., 2024a); iv) recent most popular LLM models: GPT4TS (Zhou et al., 2024), AutoTimes (Liu et al., 2024b), TimeLLM (Jin et al., 2024). To ensure a fair comparison, we adhere to the experimental settings of TimesNet (Wu et al., 2023). [1]

**Implementation details.** In the experiments, following previous works (Zhou et al., 2024; Liu et al., 2024b), we use GPT2 (Radford et al., 2019) as the LLM encoder for text-prompt representation learning. All datasets will follow a split ratio of 7:1:2 for the training, validation, and testing sets, respectively. For evaluation, we adopt the widely used metrics mean square error (MSE) and mean absolute error (MAE) (Wu et al., 2021; 2023; Nie et al., 2023b; Zhou et al., 2024). The history length $T$ is searched from the $\{96, 192, 336, 512, 672, 1024\}$ based on the best validation MSE values for all methods. Other hyperparameters, such as learning rate and network configurations for different baselines, are set based on their official code in Appendix D. In addition, channel-independence is crucial for multivariate time series prediction (Nie et al., 2023a), so it is necessary to verify the performance of models on a single channel to ensure their effectiveness across all channels in multivariate time series prediction. In this paper, experiments were conducted on a single channel as suggested by Jia et al. (2023). All experiments were conducted using PyTorch Paszke et al. (2019) on NVIDIA 3090-24G GPUs.

### 4.2 MAIN RESULTS (RQ1)

**Long-range forecasting.** In this section, we consider long-range prediction tasks on four real-world datasets: `Electricity`, `Traffic`, `ETTh1`, and `ETTm1`. As shown in Table 1, the proposed model achieves the best average performance in the long-range prediction tasks. Specifically, the proposed models consistently outperform the linear ensemble model MoLE and TimeMixer with an

---

[1]In this section, the following abbreviations are used: "TimesN." for TimesNet, "S.TSF" for SparseTSF, "T.Mixer" for TimeMixer, "MoLE" for MoLE, "Infr." for Informer, "Autofr." for Autoformer, "GPT4TS" for GPT4TS, "AutoT." for AutoTimes and "T.LLM" for TimeLLM.

average improvement of 23.17% and 20.70% respectively in terms of MSE, which demonstrates the effectiveness of using multimodal knowledge. As using a large language model for text information extraction, the proposed mixture of linear experts is allowed for better modeling of nonlinear parts in real-world time series. By comparing the LLM-based time series model GPT4TS and AutoTimes, we also have average improvements of 11.76% and 29.85% in terms of MSE. This demonstrates the effectiveness of the proposed multimodal fusion strategies and multiple linear expert ensembles. Directly aligning language models for time series may degrade the forecasting performance due to the essential differences between the time series structure and the natural language syntactic structure (Tan et al., 2024). Due to the lack of space, MAE results are reported in Appendix E.

| | $H$ | Linear-mixer | | | LLM-based | | | Linear-based | | | | Transformer-based | | | | others | |
|---|---|---|---|---|---|---|---|---|---|---|---|---|---|---|---|---|---|
| | | Ours | MoLE | T.Mixer | AutoT. | T.LLM | GPT4TS | S.TSF | FITS | DLinear | LightTS | iTrans. | PatchT. | Infr. | Autofr. | TSMixer | TimesN. |
| Electricity | 96 | 0.197 | **0.195** | 0.267 | 0.234 | 0.256 | 0.209 | 0.204 | 0.200 | 0.197 | 0.247 | 0.254 | 0.312 | 0.268 | 0.595 | 0.322 | 0.278 |
| | 192 | **0.217** | 0.228 | 0.287 | 0.321 | 0.302 | 0.250 | 0.236 | 0.235 | 0.229 | 0.285 | 0.307 | 0.355 | 0.280 | 0.515 | 0.332 | 0.290 |
| | 336 | **0.241** | 0.262 | 0.466 | 0.383 | 0.467 | 0.289 | 0.268 | 0.270 | 0.263 | 0.323 | 0.358 | 0.415 | 0.332 | 0.539 | 0.377 | 0.341 |
| | 720 | **0.255** | 0.299 | 0.392 | 0.276 | 0.448 | 0.381 | 0.315 | 0.323 | 0.297 | 0.364 | 0.395 | 0.477 | 0.615 | 0.627 | 0.429 | 0.415 |
| | Avg | **0.227** | 0.246 | 0.353 | 0.304 | 0.405 | 0.282 | 0.256 | 0.257 | 0.246 | 0.305 | 0.328 | 0.390 | 0.374 | 0.569 | 0.365 | 0.331 |
| Traffic | 96 | **0.112** | 0.123 | 0.152 | 0.278 | 0.145 | 0.136 | 0.116 | 0.117 | 0.135 | 0.233 | 0.274 | 0.133 | 0.218 | 0.243 | 0.170 | 0.158 |
| | 192 | **0.117** | 0.124 | 0.147 | 0.280 | 0.145 | 0.137 | 0.118 | 0.128 | 0.137 | 0.246 | 0.207 | 0.137 | 0.259 | 0.235 | 0.176 | 0.148 |
| | 336 | **0.113** | 0.123 | 0.146 | 0.278 | 0.144 | 0.135 | 0.117 | 0.155 | 0.137 | 0.255 | 0.329 | 0.140 | 0.272 | 0.232 | 0.172 | 0.155 |
| | 720 | **0.117** | 0.140 | 0.166 | 0.292 | 0.168 | 0.151 | 0.132 | 0.314 | 0.154 | 0.306 | 0.236 | 0.168 | 0.319 | 0.237 | 0.203 | 0.161 |
| | Avg | **0.115** | 0.128 | 0.153 | 0.282 | 0.151 | 0.140 | 0.121 | 0.178 | 0.141 | 0.260 | 0.262 | 0.144 | 0.267 | 0.237 | 0.180 | 0.156 |
| ETTh1 | 96 | **0.052** | 0.063 | 0.056 | 0.069 | 0.063 | 0.057 | 0.063 | 0.059 | 0.062 | 0.082 | 0.065 | 0.055 | 0.149 | 0.089 | 0.155 | 0.058 |
| | 192 | **0.066** | 0.087 | 0.073 | 0.078 | 0.071 | 0.073 | 0.078 | 0.075 | 0.079 | 0.102 | **0.066** | 0.071 | 0.436 | 0.101 | 0.186 | 0.067 |
| | 336 | 0.079 | 0.107 | 0.085 | 0.085 | 0.089 | 0.087 | 0.088 | 0.086 | 0.102 | 0.123 | **0.072** | 0.083 | 0.238 | 0.117 | 0.263 | 0.084 |
| | 720 | 0.080 | 0.197 | **0.075** | 0.114 | 0.095 | 0.089 | 0.103 | 0.105 | 0.201 | 0.211 | **0.072** | 0.082 | 0.253 | 0.118 | 0.298 | 0.091 |
| | Avg | **0.069** | 0.114 | 0.072 | 0.086 | 0.079 | 0.077 | 0.083 | 0.081 | 0.111 | 0.129 | **0.069** | 0.073 | 0.269 | 0.106 | 0.225 | 0.075 |
| ETTm1 | 96 | **0.026** | 0.028 | 0.028 | 0.033 | 0.033 | **0.026** | 0.027 | 0.027 | 0.027 | 0.081 | 0.029 | 0.028 | 0.092 | 0.063 | 0.057 | 0.028 |
| | 192 | **0.039** | 0.048 | 0.046 | 0.048 | 0.048 | 0.040 | 0.040 | 0.040 | 0.042 | 0.184 | 0.045 | 0.041 | 0.227 | 0.068 | 0.163 | 0.044 |
| | 336 | **0.051** | 0.056 | 0.076 | 0.064 | 0.056 | 0.052 | 0.052 | 0.054 | 0.057 | 0.271 | 0.060 | 0.056 | 0.227 | 0.077 | 0.240 | 0.059 |
| | 720 | 0.072 | 0.075 | 0.083 | 0.080 | 0.077 | **0.070** | 0.071 | 0.071 | 0.072 | 0.368 | 0.078 | 0.074 | 0.319 | 0.112 | 0.295 | 0.081 |
| | Avg | **0.047** | 0.052 | 0.058 | 0.056 | 0.053 | **0.047** | 0.048 | 0.048 | 0.049 | 0.226 | 0.053 | 0.050 | 0.216 | 0.080 | 0.189 | 0.053 |
| All Avg | | **0.115** | 0.135 | 0.159 | 0.182 | 0.172 | 0.137 | 0.127 | 0.141 | 0.137 | 0.230 | 0.178 | 0.164 | 0.282 | 0.248 | 0.240 | 0.154 |
| $1^{st}$ Count | | 16 | 1 | 0 | 0 | 0 | 3 | 0 | 0 | 0 | 0 | 4 | 0 | 0 | 0 | 0 | 0 |

Table 2: MSE results of long-range forecasting. A lower value indicates better performance. The best results are highlighted in bold. The second best is underlined.

**Few-shot forecasting** refers to the scenario of making predictions with limited data, which is particularly difficult for data-driven deep learning methods. Recently, LLM time series models Jin et al. (2024); Zhou et al. (2024) have shown impressive few-shot learning capabilities. In this section, we will evaluate whether the proposed multimodal time series fusion mechanism outperforms those LLM-alignment methods in forecasting tasks. We will follow the setups in (Zhou et al., 2024; Jin et al., 2024) for fair comparisons, and we will assess scenarios with limited training data (i.e., using only 10% of the training data, while keeping the test data the same for the long-range forecasting task). Table 3 summarizes the MSE results for few-shot forecasting (MAE results are left in Appendix E due to the limit of space). As can be seen, the proposed model still outperforms all other baselines regarding average performance, especially for those LLM-based prediction models. This suggests that when dealing with limited forecasting, utilizing the proposed multimodal fusion mechanism (which combines information from global and local text prompts) is a better choice than aligning large language models for time series modeling.

## 4.3 COMPONENT ANALYSIS (RQ2)

This section explores the impact of the static and dynamic prompts in LeMoLE. We analyze the effects of removing each prompt individually, as well as both prompts, on long-range forecasting and few-shot forecasting tasks. Through this experiment, we aim to provide a detailed discussion on whether and which text prompts improve prediction performance.

The results in Table 4 summarize the analysis of the components. It is evident that the prediction performance declines when either or both components are removed from the proposed LeMoLE. This shows that introducing the text modality using the proposed multimodality fusion strategy is effective. Interestingly, we observed that in the non-stationary ETT datasets, the proposed LeMoLE benefits more from the dynamic prompt. On the other hand, for ECL, which is relatively easy due to

| | H | Linear-mixer | | | LLM-based | | | Linear-based | | | | Transformer-based | | | | others | |
|---|---|---|---|---|---|---|---|---|---|---|---|---|---|---|---|---|---|
| | | Ours | MoLE | T.Mixer | AutoT. | T.LLM | GPT4TS | S.TSF | FITS | DLinear | LightTS | iTrans. | PatchT. | Infr. | Autofr. | TSMixer | TimesN. |
| Electricity | 96 | **0.263** | 0.276 | 0.307 | 0.505 | 0.298 | 0.304 | 0.275 | 0.397 | 0.362 | 0.508 | 0.336 | 0.354 | 0.937 | 0.691 | 0.399 | 0.348 |
| | 192 | 0.307 | **0.298** | 0.350 | 0.527 | 0.312 | 0.323 | 0.351 | 0.629 | 0.416 | 0.515 | 0.385 | 0.365 | 0.896 | 0.599 | 0.437 | 0.382 |
| | 336 | 0.337 | **0.323** | 0.374 | 0.553 | 0.328 | 0.354 | 0.391 | 0.740 | 0.443 | 0.563 | 0.399 | 0.442 | 1.264 | 0.751 | 0.508 | 0.457 |
| | 720 | 0.437 | 0.457 | 0.488 | 0.642 | 0.443 | 0.506 | **0.417** | 1.037 | 0.547 | 0.676 | 0.542 | 0.505 | 1.243 | 0.711 | 0.650 | 0.640 |
| | Avg | **0.336** | 0.339 | 0.380 | 0.557 | 0.345 | 0.372 | 0.359 | 0.701 | 0.442 | 0.565 | 0.415 | 0.417 | 1.085 | 0.688 | 0.498 | 0.457 |
| Traffic | 96 | **0.142** | 0.240 | 0.163 | 1.280 | 0.238 | 0.156 | 0.210 | 0.878 | 0.257 | 0.710 | 0.196 | 0.159 | 1.967 | 0.358 | 0.570 | 0.183 |
| | 192 | **0.153** | 0.246 | 0.180 | 1.303 | 0.241 | 0.157 | 0.227 | 1.457 | 0.257 | 0.683 | 0.194 | 0.161 | 1.333 | 0.501 | 0.521 | 0.212 |
| | 336 | **0.155** | 0.254 | 0.171 | 1.328 | 0.321 | 0.165 | 0.245 | 1.645 | 0.262 | 0.655 | 0.181 | 0.161 | 1.872 | 0.380 | 0.560 | 0.217 |
| | 720 | **0.187** | 0.322 | 0.215 | 1.431 | 0.357 | 0.204 | 0.419 | 2.377 | 0.307 | 0.867 | 0.240 | 0.189 | 1.953 | 0.465 | 0.571 | 0.330 |
| | Avg | **0.159** | 0.265 | 0.182 | 1.336 | 0.289 | 0.170 | 0.275 | 1.589 | 0.271 | 0.729 | 0.203 | 0.167 | 1.781 | 0.426 | 0.555 | 0.236 |
| ETTh1 | 96 | 0.065 | 0.072 | 0.068 | 0.381 | 0.073 | 0.070 | 0.074 | 0.074 | 0.074 | 1.273 | 0.062 | **0.060** | 1.926 | 0.304 | 1.908 | 0.073 |
| | 192 | **0.071** | 0.086 | 0.087 | 0.503 | 0.108 | 0.085 | 0.090 | 0.091 | 0.089 | 1.566 | 0.088 | 0.094 | 2.695 | 0.349 | 1.258 | 0.093 |
| | 336 | **0.074** | 0.093 | 0.116 | 0.831 | 0.150 | 0.087 | 0.112 | 0.103 | 0.123 | 1.729 | 0.106 | 0.265 | 3.398 | 0.338 | 1.288 | 0.179 |
| | 720 | **0.083** | 0.207 | 0.102 | 6.660 | 0.227 | 0.114 | 0.154 | 0.154 | 0.097 | 2.170 | 0.119 | 0.280 | 7.022 | 0.720 | 2.032 | 0.171 |
| | Avg | **0.073** | 0.115 | 0.093 | 2.094 | 0.139 | 0.089 | 0.108 | 0.106 | 0.096 | 1.684 | 0.094 | 0.175 | 3.760 | 0.428 | 1.621 | 0.129 |
| ETTm1 | 96 | **0.030** | 0.037 | 0.041 | 0.063 | 0.048 | 0.031 | 0.032 | 0.038 | 0.037 | 1.175 | 0.032 | 0.039 | 5.233 | 0.345 | 2.023 | 0.033 |
| | 192 | **0.043** | 0.049 | 0.047 | 0.073 | 0.055 | 0.044 | 0.044 | 0.050 | 0.055 | 1.356 | 0.047 | 0.060 | 6.433 | 1.263 | 1.515 | 0.049 |
| | 336 | **0.054** | 0.063 | 0.062 | 0.083 | 0.062 | **0.054** | 0.057 | 0.060 | 0.067 | 1.602 | 0.062 | 0.067 | 5.837 | 5.759 | 1.484 | 0.064 |
| | 720 | 0.081 | 0.085 | 0.093 | 0.103 | 0.100 | 0.085 | 0.079 | **0.078** | 0.083 | 1.698 | 0.086 | 0.126 | 7.920 | 15.005 | 1.847 | 0.093 |
| | Avg | **0.052** | 0.059 | 0.060 | 0.081 | 0.066 | 0.054 | 0.053 | 0.057 | 0.060 | 1.458 | 0.057 | 0.073 | 6.356 | 5.593 | 1.717 | 0.060 |
| All Avg | | **0.155** | 0.231 | 0.179 | 1.017 | 0.210 | 0.520 | 0.199 | 0.613 | 0.217 | 1.109 | 0.192 | 0.208 | 3.246 | 1.784 | 1.098 | 0.220 |
| $1^{st}$ Count | | 15 | 2 | 0 | 0 | 0 | 1 | 1 | 1 | 0 | 0 | 0 | 1 | 0 | 0 | 0 | 0 |

Table 3: MSE results for few-shot case on 10% of training data. Lower is better, with the best results highlighted in bold and the second best underlined.

its significant periodicity, the dynamic prompt is less important than the static prompt. This could be explained by the fact that the dynamic prompt introduces more local temporal information suitable for capturing non-stationary temporal behaviors. When a forecasting task exhibits significant periodic behaviors, the static prompt with global information contributes relatively more.

| Tasks | Long-range forecasting | | | | Few-shot forecasting | | | |
|---|---|---|---|---|---|---|---|---|
| Dataset | ETTh | | Electricity | | ETT | | Electricity | |
| | MSE | ↓ | MSE | ↓ | MSE | ↓ | MSE | ↓ |
| Ours | 0.0527 | - | 0.241 | - | 0.0643 | - | 0.338 | - |
| w/o Static Prompt | 0.0530 | 0.57% | 0.296 | 23.00% | 0.0756 | 17.57% | 0.357 | 5.40% |
| w/o Dynamic Prompt | 0.0536 | 1.71% | 0.276 | 14.89% | 0.0764 | 18.82% | 0.348 | 2.86% |
| w/o Both Prompts | 0.0538 | 2.09% | 0.328 | 36.22% | 0.0772 | 20.06% | 0.387 | 14.18% |

Table 4: Ablations study of the proposed model design in predicting 336 steps on `ETTh1`, `ETTm1` and `Electricity`. A lower value indicates better performance. The best results are highlighted in bold. The second best is underlined. ↓ indicates the degradation percentage.

### 4.4 MIXUP OF LINEAR EXPERTS IN TIME OR FREQUENCY DOMAIN (RQ3)

This section compares the proposed LeMoLE-T with its frequency extension LeMoLE-F introduced in Section 3. In Figure 3, the MSE prediction errors are reported with varying horizon $H$'s. As can be seen, the mixture of time experts proves to be a better choice than that of the frequency experts in the proposed LeMoLE framework. This is mainly because LeMoLE-T contains experts with different historical lookback lengths, allowing for good short- and long-term pattern modeling. On the other hand, LeMoLE-F is based on the linear frequency model FITS (Xu et al., 2024), which emphasizes modeling low-frequency components and tends to generate smooth trends while overlooking detailed local variations.

### 4.5 EFFECTS OF THE NUMBER OF EXPERTS (RQ4)

In this experiment, we analyze the effects of the number of experts in the proposed LeMoLE. In Figure 4, we observed that when the prediction task is relative stationary and with significant periodic, say `Electricity` and `Traffic`, the number of experts for mixture is relatively small.

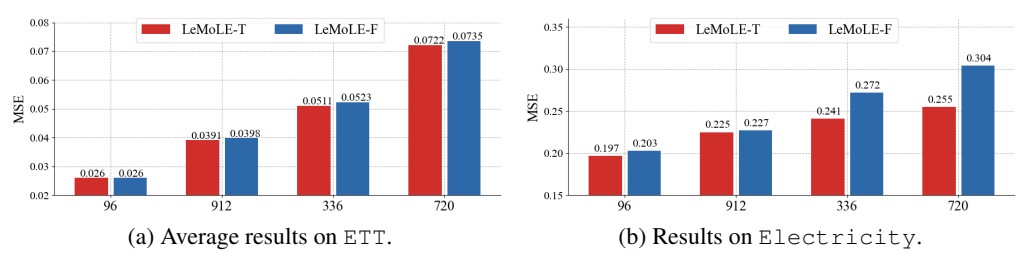

Figure 3: MSE resulrs of time vs. frequency experts.

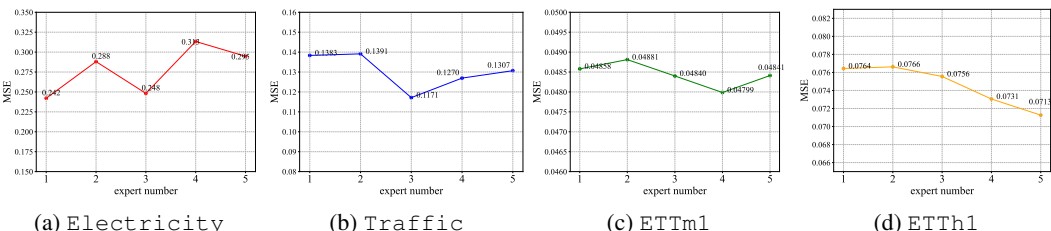

Figure 4: Average MSE's with varying numbers of experts in LeMoLE.

For example, the best number of experts on `Electricity` and `Traffic` are 1 and 3, respectively. However, more experts are expected for more challenging datasets `ETTh1` and `ETTm1` that are highly nonlinear and non-stationary.

### 4.6 EFFICIENCY (RQ5)

Table 5 shows the number of trainable parameters and inference speed. The existing alignment-based LLM models suffer from slower training and inference speeds due to the immensity of LLMs. While AutoTimes has a faster inference speed compared to TimeLLM due to its patching-based inference strategy, its autoregressive decoding process still necessitates multiple forward processes of LLM. The inference efficiency of LeMoLE over existing LLM time series models is due to: i) In LeMoLE, time series are modeled using a combination of linear experts instead of aligning a large language model with time series. This results in lower computational costs. ii) Additionally, the multimodal fusion module is implemented using lightweight CNNs, avoiding the introduction of additional self-attention layers, which have quadratic complexity with the length of the time series for time series-text alignment.

| Metric | *H*=96 | | | *H*=720 | | |
|---|---|---|---|---|---|---|
| Unit | Param. (M) | Train. (ms) | Inference. (ms) | Param. (M) | Train. (ms) | Inference. (ms) |
| DLinear | 0.098 | 0.032 | 0.325 | 0.277 | 0.036 | 0.244 |
| MoLE | 0.493 | 0.061 | 0.404 | 0.738 | 0.071 | 0.464 |
| TimeMixer | 0.075 | 0.589 | 0.574 | 0.190 | 0.633 | 4.541 |
| AutoTimes | 0.148 | 15.22 | 8.781 | 0.148 | 16.96 | 68.86 |
| TimeLLM | 53.44 | 1.740 | 22.312 | 58.55 | 1.772 | 22.87 |
| GPT4TS | 3.920 | 0.329 | 3.964 | 24.04 | 0.398 | 4.030 |
| LeMoLE-T | 0.514 | 0.163 | 1.209 | 3.850 | 0.197 | 1.306 |
| LeMoLE-F | 0.431 | 0.194 | 2.219 | 3.030 | 0.352 | 2.865 |

Table 5: Efficiency analysis: number of trainable parameters and training/inference speed (in s) of various time series models.

## 5 CONCLUSION

This study introduces LeMoLE, a multimodal mixture of linear experts, for time series forecasting. By harnessing the powerful capabilities of a pre-trained large language model, LeMoLE allows for a flexible ensemble of multiple linear experts by integrating static and dynamic text knowledge correlated to time series data. By comparing existing LLM time series models aligning text and time series in large language models' spaces, the proposed LeMoLE shows greater effectiveness. This finding demonstrates the effectiveness of a mixture of linear experts and the use of multimodal knowledge. Furthermore, the study delves into detailed discussions regarding the variant of frequency experts and computational costs.

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

## A  PSEUDO-CODE OF TRAINING PROCEDURE.

The training procedure of LeMoLE is shown in Algorithms 1.

---

**Algorithm 1** Training procedure for LeMoLE

---

1: **repeat**
2:     **Input:** Time series $\mathbf{X}_{1:T} \in \mathbb{R}^{T \times C}$, static prompt $\mathbf{P}^s$
3:     **Initialization:** Learning rate $\eta$, number of experts $M$, window lengths $\{w_1, w_2, \ldots, w_M\}$
4:     **for** $m = 1, 2, \ldots, M$ **do**
5:         Transform $\mathbf{X}_{1:T}$ into sub-series $\mathbf{X}_{T-w_m:T}$ for the $m$-th expert
6:         Generate dynamic prompt $\mathbf{P}_m^d$ for the $m$-th expert
7:         Obtain prediction $\hat{\mathbf{Y}}_{T+1:T+H}^{(m)} = \mathbf{W}_m \mathbf{X}_{T-w_m:T} + \boldsymbol{b}_m$, see Equation (3)
8:     **end for**
9:     Encode static prompt $\mathbf{P}^s$ and dynamic prompts $\{\mathbf{P}_m^d\}_{m=1}^M$ using LLM to get $\boldsymbol{Z}_S$ and $\boldsymbol{Z}_D^{(m)}$
10:    **for** $m = 1, 2, \ldots, M$ **do**
11:       Fuse static representation $\boldsymbol{Z}_S$ with expert outputs $\hat{\mathbf{Y}}_{T+1:T+H}^{(m)}$ using FiLM: $\boldsymbol{\gamma}_S^{(m)} = \text{Linear}_{S,1}(\boldsymbol{Z}_S), \boldsymbol{\beta}_S^{(m)} = \text{Linear}_{S,2}(\boldsymbol{Z}_S)$
12:       Apply FiLM Layer: $\hat{\mathbf{Y}}_{T+1:T+H}^{(m)'} = \boldsymbol{\gamma}_S^{(m)} \odot \hat{\mathbf{Y}}_{T+1:T+H}^{(m)} + \boldsymbol{\beta}_S^{(m)}$, see Equation ( 6)
13:       Fuse dynamic representation $\boldsymbol{Z}_D^{(m)}$: $\boldsymbol{\gamma}_D^{(m)} = \text{Linear}_{D,1}(\boldsymbol{Z}_D^{(m)}), \boldsymbol{\beta}_D^{(m)} = \text{Linear}_{D,2}(\boldsymbol{Z}_D^{(m)})$
14:       Apply FiLM Layer: $\hat{\mathbf{Y}}_{T+1:T+H}^{(m)''} = \boldsymbol{\gamma}_D^{(m)} \odot \hat{\mathbf{Y}}_{T+1:T+H}^{(m)'} + \boldsymbol{\beta}_D^{(m)}$, see Equation ( 7)
15:    **end for**
16:    **Ensemble Output:** $\hat{\mathbf{Y}}_{T+1:T+H} = \text{CNN}_{\text{final}}\left([\hat{\mathbf{Y}}_{T+1:T+H}^{(1)''}, \ldots, \hat{\mathbf{Y}}_{T+1:T+H}^{(M)''}]\right)$, see Equation (8)
17:    **Loss Calculation:** $L(\theta) = ||\mathbf{X}_{T+1:T+H} - \hat{\mathbf{Y}}_{T+1:T+H}||_2^2$, see ( 9)
18:    **Gradient Update:** $\theta \leftarrow \theta - \eta \nabla_\theta L(\theta)$
19: **until** converged

---

## B  PROMPT EXAMPLE.

In this section, we provide a prompt example regarding the static and dynamic text prompts used in the proposed model. Figure 5 shows the text prompts on the `ETT` dataset.

| Static Prompt | Dynamic Prompt |
|---|---|
| The Electricity Transformer Temperature (ETT) is a crucial indicator in the electric power long-term deployment. This dataset consists of 2 years data from two separated counties in China. To explore the granularity on the Long sequence time-series forecasting (LSTF) problem, different subsets are created, {ETTh1, ETTh2} for 1-hour-level and ETTm1 for 15-minutes-level. Each data point consists of the target value "oil temperature" and 6 power load features. The train/val/test is 12/4/4 months. | \<Format\> This is the series including \<start timestamp\>,..., \<end timestamp\>. |

Figure 5: Text prompt examples on `ETT` dataset.

## C  SUPPLEMENTARY OF DATASETS

| Dataset | Sampling Frequency | Total Observations | Dimension |
|---|---|---|---|
| Electricity | 1 hour | 26,304 | 321 |
| Traffic | 1 hour | 17,544 | 862 |
| ETTh1 | 1 hour | 17,544 | 7 |
| ETTm1 | 1 min | 69,680 | 7 |

Table 6: Summary of datasets

The `ETT` (Electricity Transformer Temperature) [2] (Zhou et al., 2021) encompasses a comprehensive collection of transformer operational data, consisting of two subsets: `ETTh`, featuring hourly recordings, and `ETTm`, with data collected at a finer 15-minute interval. Both subsets span the period from July 2016 to July 2018. The `Traffic` [3] provides insights into road congestion patterns by detailing occupancy rates along San Francisco's freeway network, and encompasses hourly measurements spanning from 2015 through 2016. The `Electricity` [4] compiles hourly records of energy usage from a cohort of 321 individual clients, spanning a three-year time-frame between 2012 and 2014.

## D    SUPPLEMENTARY OF BASELINES

Recent transformer variants aim to improve the standard transformer structure for time series modeling (Wen et al., 2022; Zhou et al., 2021; Wu et al., 2023). For example, to reduce the time complexity and memory usage, Informer (Wen et al., 2022) proposes ProbSparse self-attention mechanism and the adoption of a generative decoder to reduce the time complexity and memory usage. Autoformer (Wu et al., 2021) adopts data decomposition techniques and designs an efficient auto-correlation mechanism to improve prediction accuracy. To analyze time series in the multi-scale aspect, Pyraformer (Liu et al., 2022a) implements intra-scale and inter-scale attention to capture temporal dependencies across different resolutions effectively. In the frequency domain, FEDFormer (Zhou et al., 2022a) designs the enhanced blocks with Fourier transform and wavelet transform, enabling the focus on capturing important structures in time series through frequency domain mapping. Recently, PatchTST (Nie et al., 2023a) segments time series into patches that serve as input tokens to Transformer and use the channel independence assumption to get better performance. To capture the relationship between the variables, iTransformer (Liu et al., 2024a) replaces the standard attention across the time with variable attention while keeping the whole structure of the standard transformer model.

MoLE: https://github.com/RogerNi/MoLE; TimeMixer: https://github.com/kwuking/TimeMixer; TSMixer: https://github.com/google-research/google-research/tree/master/tsmixer; AutoTimes: https://github.com/thuml/AutoTimes; Time-LLM: https://github.com/KimMeen/Time-LLM; GPT4TS https://github.com/DAMO-DI-ML/NeurIPS2023-One-Fits-All; SparseTSF: https://github.com/lss-1138/SparseTSF; FITS: https://github.com/VEWOXIC/FITS;DLinear: https://github.com/cure-lab/LTSF-Linear LightTS: https://tinyurl.com/5993cmus; iTransformer: https://github.com/thuml/iTransformer; PatchTST: https://github.com/yuqinie98/PatchTST; Informer: https://github.com/zhouhaoyi/Informer2020; Autoformer: https://github.com/thuml/Autoformer; TimesNet: https://github.com/thuml/Time-Series-Library

---

[2]https://github.com/zhouhaoyi/ETDataset

[3]https://pems.dot.ca.gov/

[4]https://archive.ics.uci.edu/dataset/321/electricityloaddiagrams20112014

# E LONG-RANGE AND FEW SHOT FORECASTING (RQ1&RQ2)

Mean absolute error (MAE) is another important metric in time series forecasting tasks. We list the MAE results in Table 7 and Table 8 following the same experiment environments in RQ1 and RQ2.

| | H | Linear-mixer | | | LLM-based | | | Linear-based | | | | Transformer-based | | | | others | |
| | | Ours | MoLE | T.Mixer | AutoT. | T.LLM | GPT4TS | S.TSF | FITS | DLinear | LightTS | iTrans. | PatchT. | Infr. | Autofr. | TSMixer | TimesN. |
|---|---|---|---|---|---|---|---|---|---|---|---|---|---|---|---|---|---|
| Electricity | 96 | 0.311 | **0.307** | 0.377 | 0.351 | 0.353 | 0.318 | 0.312 | 0.309 | 0.309 | 0.359 | 0.363 | 0.411 | 0.373 | 0.566 | 0.404 | 0.384 |
| | 192 | 0.335 | **0.330** | 0.378 | 0.413 | 0.380 | 0.348 | 0.335 | 0.334 | 0.331 | 0.382 | 0.400 | 0.418 | 0.380 | 0.522 | 0.411 | 0.388 |
| | 336 | **0.353** | 0.358 | 0.470 | 0.461 | 0.486 | 0.376 | 0.360 | 0.364 | 0.359 | 0.409 | 0.435 | 0.448 | 0.421 | 0.547 | 0.441 | 0.419 |
| | 720 | **0.375** | 0.402 | 0.454 | 0.381 | 0.478 | 0.443 | 0.411 | 0.423 | 0.401 | 0.448 | 0.463 | 0.507 | 0.601 | 0.586 | 0.483 | 0.466 |
| | Avg | **0.344** | 0.349 | 0.420 | 0.401 | 0.424 | 0.371 | 0.355 | 0.357 | 0.350 | 0.400 | 0.415 | 0.446 | 0.444 | 0.555 | 0.435 | 0.414 |
| Traffic | 96 | 0.189 | 0.204 | 0.245 | 0.378 | 0.224 | 0.229 | **0.185** | 0.193 | 0.228 | 0.335 | 0.370 | 0.205 | 0.312 | 0.348 | 0.187 | 0.242 |
| | 192 | 0.193 | 0.206 | 0.233 | 0.379 | 0.228 | 0.229 | **0.187** | 0.211 | 0.230 | 0.348 | 0.313 | 0.211 | 0.339 | 0.340 | 0.329 | 0.235 |
| | 336 | 0.191 | 0.208 | 0.244 | 0.379 | 0.230 | 0.230 | **0.190** | 0.263 | 0.233 | 0.359 | 0.420 | 0.218 | 0.361 | 0.340 | 0.409 | 0.248 |
| | 720 | **0.200** | 0.232 | 0.273 | 0.388 | 0.262 | 0.247 | 0.208 | 0.439 | 0.257 | 0.398 | 0.338 | 0.247 | 0.392 | 0.345 | 0.474 | 0.259 |
| | Avg | 0.194 | 0.212 | 0.249 | 0.381 | 0.236 | 0.234 | **0.193** | 0.277 | 0.237 | 0.360 | 0.360 | 0.220 | 0.351 | 0.343 | 0.349 | 0.246 |
| ETTh1 | 96 | **0.175** | 0.192 | 0.181 | 0.203 | 0.197 | 0.186 | 0.197 | 0.188 | 0.186 | 0.219 | 0.197 | 0.181 | 0.315 | 0.239 | 0.327 | 0.187 |
| | 192 | 0.203 | 0.227 | 0.207 | 0.219 | 0.210 | 0.212 | 0.220 | 0.213 | 0.212 | 0.244 | 0.203 | 0.209 | 0.592 | 0.244 | 0.360 | **0.199** |
| | 336 | **0.223** | 0.257 | 0.225 | 0.231 | 0.237 | 0.235 | 0.238 | 0.234 | 0.249 | 0.275 | **0.213** | 0.227 | 0.416 | 0.270 | 0.443 | 0.223 |
| | 720 | 0.233 | 0.367 | 0.221 | 0.266 | 0.243 | 0.236 | 0.252 | 0.256 | 0.370 | 0.383 | **0.217** | 0.227 | 0.428 | 0.270 | 0.478 | 0.239 |
| | Avg | **0.208** | 0.261 | 0.208 | 0.230 | 0.222 | 0.217 | 0.227 | 0.223 | 0.254 | 0.280 | **0.207** | 0.211 | 0.438 | 0.256 | 0.402 | 0.212 |
| ETTm1 | 96 | **0.123** | 0.124 | 0.126 | 0.140 | 0.142 | 0.124 | 0.124 | 0.127 | 0.124 | 0.225 | 0.129 | 0.127 | 0.247 | 0.191 | 0.187 | 0.126 |
| | 192 | **0.151** | 0.163 | 0.161 | 0.168 | 0.170 | 0.152 | **0.151** | 0.153 | 0.153 | 0.340 | 0.163 | 0.156 | 0.411 | 0.205 | 0.329 | 0.159 |
| | 336 | **0.172** | 0.177 | 0.215 | 0.192 | 0.181 | 0.174 | 0.174 | 0.177 | 0.178 | 0.441 | 0.188 | 0.183 | 0.401 | 0.219 | 0.409 | 0.186 |
| | 720 | 0.206 | 0.205 | 0.221 | 0.216 | 0.215 | 0.204 | **0.203** | 0.205 | 0.204 | 0.522 | 0.215 | 0.209 | 0.474 | 0.271 | 0.474 | 0.218 |
| | Avg | **0.163** | 0.167 | 0.181 | 0.179 | 0.177 | **0.163** | **0.163** | 0.166 | 0.165 | 0.382 | 0.174 | 0.169 | 0.383 | 0.221 | 0.349 | 0.172 |
| 1^st Count | | 10 | 2 | 0 | 0 | 0 | 1 | 7 | 0 | 0 | 0 | 3 | 0 | 0 | 0 | 0 | 1 |
| All Avg | | **0.227** | 0.247 | 0.265 | 0.298 | 0.265 | 0.246 | 0.235 | 0.256 | 0.252 | 0.356 | 0.289 | 0.262 | 0.404 | 0.344 | 0.384 | 0.261 |

Table 7: Full long-term forecasting MAE results of univariate time series. We set the forecasting horizons $H \in \{96, 192, 336, 720\}$ for all datasets. A lower value indicates better performance. The best results are highlighted in bold. The second best is underlined.

| | H | Linear-mixer | | | LLM-based | | | Linear-based | | | | Transformer-based | | | | others | |
| | | Ours | MoLE | T.Mixer | AutoT. | T.LLM | GPT4TS | S.TSF | FITS | DLinear | LightTS | iTrans. | PatchT. | Infr. | Autofr. | TSMixer | TimesN. |
|---|---|---|---|---|---|---|---|---|---|---|---|---|---|---|---|---|---|
| Electricity | 96 | **0.369** | 0.385 | 0.399 | 0.539 | 0.401 | 0.408 | 0.371 | 0.470 | 0.466 | 0.551 | 0.412 | 0.436 | 0.715 | 0.666 | 0.466 | 0.430 |
| | 192 | 0.407 | **0.403** | 0.431 | 0.549 | 0.410 | 0.422 | 0.420 | 0.614 | 0.506 | 0.552 | 0.446 | 0.439 | 0.717 | 0.591 | 0.495 | 0.454 |
| | 336 | **0.423** | 0.425 | 0.451 | 0.563 | 0.428 | 0.444 | 0.449 | 0.672 | 0.528 | 0.579 | 0.462 | 0.490 | 0.846 | 0.662 | 0.542 | 0.508 |
| | 720 | 0.498 | 0.527 | 0.524 | 0.618 | 0.516 | 0.554 | **0.483** | 0.814 | 0.597 | 0.635 | 0.564 | 0.539 | 0.843 | 0.644 | 0.619 | 0.610 |
| | Avg | **0.425** | 0.435 | 0.451 | 0.567 | 0.439 | 0.457 | 0.431 | 0.643 | 0.524 | 0.579 | 0.471 | 0.476 | 0.780 | 0.641 | 0.531 | 0.501 |
| Traffic | 96 | **0.238** | 0.341 | 0.260 | 0.948 | 0.348 | 0.250 | 0.304 | 0.782 | 0.360 | 0.654 | 0.301 | 0.247 | 1.111 | 0.456 | 0.615 | 0.285 |
| | 192 | 0.254 | 0.346 | 0.280 | 0.958 | 0.351 | **0.250** | 0.317 | 1.016 | 0.360 | 0.637 | 0.300 | 0.255 | 0.910 | 0.563 | 0.573 | 0.315 |
| | 336 | **0.250** | 0.356 | 0.275 | 0.969 | 0.421 | 0.261 | 0.334 | 1.077 | 0.369 | 0.618 | 0.290 | 0.259 | 1.118 | 0.480 | 0.602 | 0.332 |
| | 720 | **0.290** | 0.406 | 0.323 | 1.011 | 0.453 | 0.303 | 0.467 | 1.281 | 0.401 | 0.707 | 0.352 | 0.291 | 1.179 | 0.518 | 0.564 | 0.427 |
| | Avg | **0.258** | 0.362 | 0.284 | 0.972 | 0.393 | 0.266 | 0.356 | 1.039 | 0.372 | 0.654 | 0.311 | 0.263 | 1.080 | 0.504 | 0.588 | 0.339 |
| ETTh1 | 96 | 0.200 | 0.202 | 0.201 | 0.459 | 0.213 | 0.204 | 0.213 | 0.212 | 0.211 | 0.933 | 0.190 | **0.186** | 1.335 | 0.428 | 1.191 | 0.208 |
| | 192 | **0.214** | 0.227 | 0.233 | 0.519 | 0.254 | 0.229 | 0.237 | 0.237 | 0.234 | 1.027 | 0.229 | 0.234 | 1.551 | 0.457 | 0.972 | 0.233 |
| | 336 | **0.217** | 0.244 | 0.271 | 0.640 | 0.317 | 0.234 | 0.271 | 0.255 | 0.282 | 1.082 | 0.258 | 0.409 | 1.780 | 0.442 | 0.915 | 0.345 |
| | 720 | **0.232** | 0.369 | 0.253 | 1.469 | 0.394 | 0.268 | 0.317 | 0.313 | 0.251 | 1.235 | 0.273 | 0.442 | 2.604 | 0.714 | 1.148 | 0.332 |
| | Avg | **0.216** | 0.261 | 0.240 | 0.772 | 0.295 | 0.234 | 0.260 | 0.254 | 0.244 | 1.069 | 0.238 | 0.318 | 1.818 | 0.510 | 1.056 | 0.280 |
| ETTm1 | 96 | **0.132** | 0.146 | 0.152 | 0.193 | 0.169 | 0.133 | 0.138 | 0.153 | 0.149 | 0.884 | 0.134 | 0.151 | 2.232 | 0.528 | 1.389 | 0.136 |
| | 192 | **0.158** | 0.167 | 0.167 | 0.207 | 0.179 | 0.159 | 0.161 | 0.174 | 0.177 | 0.998 | 0.166 | 0.189 | 2.486 | 1.055 | 1.178 | 0.168 |
| | 336 | **0.179** | 0.190 | 0.193 | 0.221 | 0.192 | **0.178** | 0.183 | 0.189 | 0.194 | 1.085 | 0.191 | 0.201 | 2.376 | 2.327 | 1.159 | 0.195 |
| | 720 | **0.214** | 0.223 | 0.234 | 0.249 | 0.243 | 0.222 | 0.216 | 0.217 | 0.222 | 1.091 | 0.225 | 0.277 | 2.778 | 3.766 | 1.323 | 0.234 |
| | Avg | **0.171** | 0.182 | 0.187 | 0.218 | 0.196 | 0.173 | 0.174 | 0.183 | 0.186 | 1.015 | 0.179 | 0.205 | 2.468 | 1.919 | 1.262 | 0.183 |
| All Avg | | 0.267 | 0.340 | 0.291 | 0.632 | 0.331 | 0.421 | 0.305 | 0.530 | 0.332 | 0.829 | 0.300 | 0.315 | 1.536 | 0.894 | 0.859 | 0.326 |
| 1^st Count | | 16 | 1 | 0 | 0 | 0 | 1 | 1 | 0 | 0 | 0 | 0 | 2 | 0 | 0 | 0 | 0 |

Table 8: Few-shot learning MAE results on 10% training data. A lower value indicates better performance. The best results are highlighted in bold. The second best is underlined.

## F  VISUALIZATION ANALYSIS

In this section, we provide visualization results on periodic `Electricity` data and nonstationary `ETTh1` data. Figure 6 and Figure 7 showcase the prediction results of various time series forecasting models, including SparseTSF, iTransformer and PatchTST, DLinear, MoLE, Time-LLM, GPT4TS, and the proposed LeMoLE.

As can be seen, for that relative smooth periodic `Electricity` data, LeMoLE can produce higher quality prediction. When dealing with the nonstationary `ETTh1` data, LLM models such as Time-LLM, GPT4TS and our LeMoLE all perform better than other methods. This is mainly due to the use of multimodal knowledge.

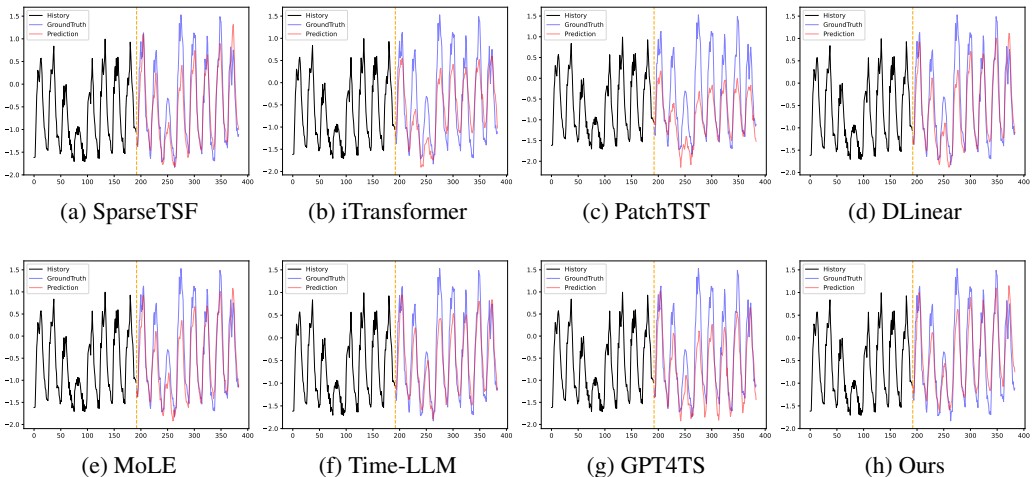

Figure 6: Prediction results on `Electricity`.

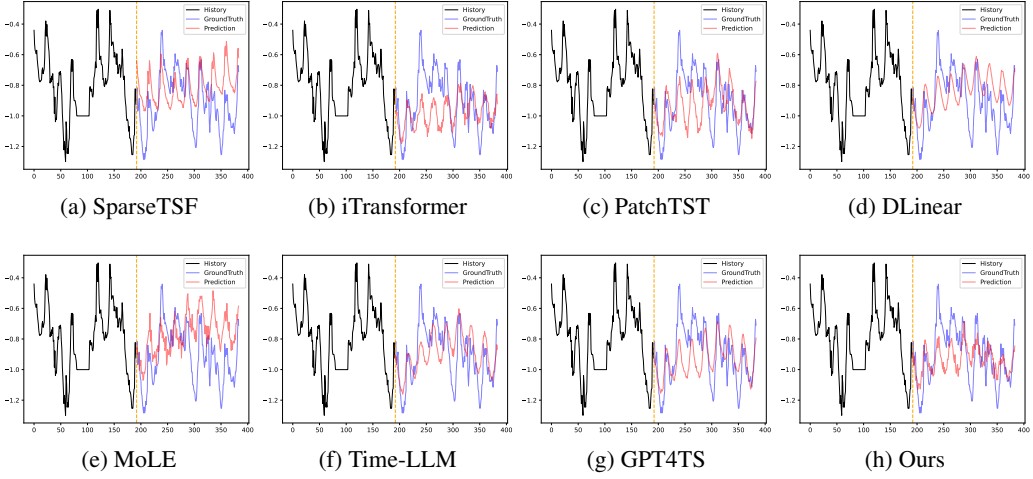

Figure 7: Prediction results on `ETTh1`.

# G    HYPERPARAMETER SENSITIVITY

Table 9 presents the results of our comparison tests between the choices of the number and type of experts. Here, we observe under the same number of experts, the temporal linear expert is better than the frequency expert in the average results.

Our analysis shows that increasing the number of experts, in the LeMoLE and LeMoLE-F models affects their performance, varying depending on the dataset as shown in Table 9. In the `Electricity` dataset, LeMoLE improves up to three experts, but additional experts add complexity without accuracy gains. In contrast, the Traffic dataset shows consistent improvements up to three experts. For the `ETTh1` and `ETTm1` datasets, stability is observed with minimal performance changes, suggesting these datasets require fewer experts. The frequency-based LeMoLE-F model benefits specific configurations but needs careful tuning for optimal results.

| Methods | | | | | | | | | | | | LeMoLE-T | | | | | | | | | LeMoLE-F | | | | | | | | | |
|---|---|---|---|---|---|---|---|---|---|---|---|---|---|---|---|---|---|---|---|---|---|---|---|---|---|---|---|---|---|---|
| num_expert | | 1 | | 2 | | 3 | | 4 | | 5 | | | 1 | | 2 | | 3 | | 4 | | 5 | |
| Metric | | MSE | MAE | MSE | MAE | MSE | MAE | MSE | MAE | MSE | MAE | | MSE | MAE | MSE | MAE | MSE | MAE | MSE | MAE | MSE | MAE |
| Electricity 96 | | 0.212 | 0.323 | 0.201 | 0.314 | 0.197 | 0.311 | 0.213 | 0.330 | 0.209 | 0.331 | | 0.208 | 0.320 | 0.213 | 0.326 | 0.203 | 0.318 | 0.227 | 0.360 | 0.207 | 0.321 |
| Electricity 192 | | 0.217 | 0.335 | 0.318 | 0.447 | 0.234 | 0.350 | 0.250 | 0.379 | 0.317 | 0.446 | | 0.240 | 0.358 | 0.230 | 0.334 | 0.227 | 0.329 | 0.251 | 0.353 | 0.243 | 0.354 |
| Electricity 336 | | 0.285 | 0.402 | 0.241 | 0.353 | 0.255 | 0.377 | 0.311 | 0.437 | 0.311 | 0.431 | | 0.271 | 0.386 | 0.272 | 0.384 | 0.368 | 0.480 | 0.326 | 0.424 | 0.286 | 0.402 |
| Electricity 720 | | 0.255 | 0.375 | 0.393 | 0.504 | 0.306 | 0.428 | 0.478 | 0.563 | 0.342 | 0.452 | | 0.549 | 0.602 | 0.304 | 0.410 | 0.308 | 0.421 | 0.386 | 0.482 | 0.336 | 0.431 |
| Avg | | **0.242** | **0.359** | 0.288 | 0.405 | 0.248 | 0.367 | 0.313 | 0.427 | 0.295 | 0.415 | | 0.317 | 0.417 | 0.255 | 0.364 | 0.276 | 0.387 | 0.297 | 0.405 | 0.268 | 0.377 |
| Traffic 96 | | 0.117 | 0.193 | 0.112 | 0.189 | 0.122 | 0.215 | 0.118 | 0.200 | 0.124 | 0.215 | | 0.124 | 0.207 | 0.135 | 0.233 | 0.139 | 0.247 | 0.149 | 0.248 | 0.131 | 0.217 |
| Traffic 192 | | 0.126 | 0.214 | 0.124 | 0.206 | 0.117 | 0.193 | 0.142 | 0.245 | 0.141 | 0.229 | | 0.117 | 0.197 | 0.140 | 0.238 | 0.130 | 0.229 | 0.119 | 0.201 | 0.127 | 0.209 |
| Traffic 336 | | 0.122 | 0.210 | 0.172 | 0.270 | 0.112 | 0.191 | 0.119 | 0.202 | 0.135 | 0.237 | | 0.156 | 0.262 | 0.136 | 0.236 | 0.134 | 0.234 | 0.144 | 0.249 | 0.116 | 0.198 |
| Traffic 720 | | 0.148 | 0.248 | 0.149 | 0.265 | 0.117 | 0.200 | 0.130 | 0.227 | 0.122 | 0.205 | | 0.150 | 0.254 | 0.149 | 0.253 | 0.151 | 0.260 | 0.166 | 0.275 | 0.155 | 0.258 |
| Avg | | 0.128 | 0.216 | 0.139 | 0.232 | **0.117** | **0.200** | 0.127 | 0.219 | 0.131 | 0.222 | | 0.137 | 0.230 | 0.140 | 0.240 | 0.139 | 0.242 | 0.144 | 0.243 | 0.132 | 0.220 |
| ETTh1 96 | | 0.062 | 0.194 | 0.061 | 0.192 | 0.056 | 0.182 | 0.053 | 0.178 | 0.052 | 0.175 | | 0.059 | 0.190 | 0.058 | 0.186 | 0.058 | 0.187 | 0.053 | 0.178 | 0.053 | 0.178 |
| ETTh1 192 | | 0.075 | 0.215 | 0.076 | 0.217 | 0.074 | 0.214 | 0.068 | 0.204 | 0.066 | 0.203 | | 0.071 | 0.209 | 0.072 | 0.210 | 0.072 | 0.212 | 0.070 | 0.205 | 0.065 | 0.201 |
| ETTh1 336 | | 0.082 | 0.230 | 0.083 | 0.230 | 0.083 | 0.229 | 0.084 | 0.231 | 0.079 | 0.225 | | 0.075 | 0.218 | 0.078 | 0.223 | 0.077 | 0.222 | 0.073 | 0.216 | 0.074 | 0.216 |
| ETTh1 720 | | 0.087 | 0.233 | 0.087 | 0.234 | 0.089 | 0.236 | 0.088 | 0.234 | 0.088 | 0.235 | | 0.097 | 0.250 | 0.086 | 0.232 | 0.092 | 0.240 | 0.086 | 0.234 | 0.084 | 0.232 |
| Avg | | 0.077 | 0.218 | 0.077 | 0.218 | 0.075 | 0.215 | 0.074 | 0.213 | **0.071** | **0.209** | | 0.076 | 0.217 | 0.073 | 0.213 | 0.075 | 0.215 | 0.071 | 0.208 | 0.069 | 0.207 |
| ETTm1 96 | | 0.027 | 0.126 | 0.027 | 0.124 | 0.027 | 0.123 | 0.026 | 0.123 | 0.027 | 0.123 | | 0.027 | 0.124 | 0.026 | 0.123 | 0.027 | 0.125 | 0.027 | 0.124 | 0.027 | 0.123 |
| ETTm1 192 | | 0.041 | 0.153 | 0.040 | 0.152 | 0.040 | 0.151 | 0.039 | 0.152 | 0.041 | 0.153 | | 0.040 | 0.152 | 0.040 | 0.151 | 0.040 | 0.153 | 0.040 | 0.152 | 0.040 | 0.151 |
| ETTm1 336 | | 0.053 | 0.175 | 0.055 | 0.177 | 0.054 | 0.178 | 0.053 | 0.175 | 0.053 | 0.174 | | 0.053 | 0.177 | 0.054 | 0.176 | 0.052 | 0.174 | 0.053 | 0.174 | 0.053 | 0.175 |
| ETTm1 720 | | 0.111 | 0.250 | 0.071 | 0.205 | 0.072 | 0.206 | 0.073 | 0.205 | 0.109 | 0.254 | | 0.078 | 0.219 | 0.074 | 0.208 | 0.077 | 0.216 | 0.075 | 0.212 | 0.074 | 0.208 |
| Avg | | 0.049 | 0.165 | 0.049 | 0.165 | 0.048 | 0.165 | **0.048** | **0.164** | 0.048 | 0.164 | | 0.049 | 0.168 | 0.048 | 0.164 | 0.049 | 0.167 | 0.049 | 0.166 | 0.048 | 0.164 |

Table 9: Comparison between the choices of the number of experts in LeMoLE(-F) and the choices of the type of experts, i.e. time or frequency
.

# H    STABILITY RESULTS

Table 10 lists both mean and STD of MSE and MAE metrics for LeMoLE with 3 runs in different random seeds on `ETTh1`, `ETTm2`, `Electricity` and `Traffic` datasets. The results show a small variance in the performance that represents the stability of our model.

| Dataset | | ETTh1 | | ETTm1 | | ECL | | Traffic | |
|---|---|---|---|---|---|---|---|---|---|
| Metric | | $MSE_{std}$ | $MAE_{std}$ | $MSE_{std}$ | $MAE_{std}$ | $MSE_{std}$ | $MAE_{std}$ | $MSE_{std}$ | $MAE_{std}$ |
| 96 | | $0.053_{0.0005}$ | $0.178_{0.0007}$ | $0.027_{0.0002}$ | $0.124_{0.0000}$ | $0.297_{0.0874}$ | $0.415_{0.0787}$ | $0.123_{0.0049}$ | $0.204_{0.0114}$ |
| 192 | | $0.066_{0.0009}$ | $0.204_{0.0009}$ | $0.040_{0.0003}$ | $0.151_{0.0004}$ | $0.238_{0.0145}$ | $0.357_{0.0175}$ | $0.123_{0.0033}$ | $0.208_{0.0072}$ |
| 336 | | $0.079_{0.0011}$ | $0.228_{0.0018}$ | $0.053_{0.0010}$ | $0.177_{0.0012}$ | $0.309_{0.0364}$ | $0.425_{0.0312}$ | $0.121_{0.0051}$ | $0.205_{0.0105}$ |
| 720 | | $0.086_{0.0010}$ | $0.233_{0.0013}$ | $0.071_{0.0011}$ | $0.205_{0.0012}$ | $0.268_{0.0151}$ | $0.392_{0.0132}$ | $0.136_{0.0538}$ | $0.223_{0.0708}$ |

Table 10: Model stability test of univariate time series with different random seeds. We report the standard error with different datasets,

