# OpenReview forum: "LeMoLE: LLM-enhanced Mixture of Linear Experts for Time Series Forecasting"
_ICLR.cc/2025/Conference — ICLR 2025 Conference Withdrawn Submission_

### Official Review · Reviewer_rnDU · 2024-10-24

**Soundness:** 2
**Presentation:** 2
**Contribution:** 2
**Rating:** 3
**Confidence:** 5

**Summary:**

This paper proposes an enhanced time series prediction method based on LLM named Lemole, which uses multiple lookback windows of different lengths to build mixture of linear experts. At the same time, the outputs by MOLE are adjusted based on the static knowledge and dynamic knowledge constructed by LLM.

**Strengths:**

1. This paper clearly describes the proposed method.
2. This paper is well organized and generally well written.

**Weaknesses:**

1. This paper is not innovative enough and the method proposed does not have many highlights compared to existing works. First of all, the mechanism based on multiple lookback windows of different lengths is not reasonable. In essence, modeling the learnable matrix $W_m$ of multiple different lookback window lengths is equivalent to learning the original linear layer parameter $W$. It can be considered that for a specific $W_m$, What it learns is the matrix $W_m^{'}$ = [$Zeros_m$, $W_m$], and $W$ can be obtained by directly combining these $W_m^{'}$ matrices. Secondly, the conditioning module is a combination of existing work. The static prompt comes from time-llm (ICLR 2024), and the dynamic prompt simply replaces the timestamp Embedding in MoLE with the output Embedding based on LLM, so there is nothing new in this module.
2. There are some problems with writing, and equation 3 is quoted incorrectly.
3. There are some doubts about the experimental results. First, the test dataset has few types. Secondly, taking the Electricity dataset as an example, the test results of this article are about twice the self-test results of PathTST (iclr 2023), and the test results of the Traffic data set are its about half, the test results of both etth1 and ettm1 datasets are not in the same order of magnitude as the test results of several existing works. It is not clear why this is the case.
4. The method proposed in this paper introduces the mixture experts based on the frequency domain is meaningless. The paper only explains why its effect is not good. If the mixture experts based on the time domain or frequency domain has its own adaptation scenarios, and analyzes it would make more sense, but based on the current description, I think his introduction is pointless.
5. In the ablation experiment on the number of experts, each data set has different sensitivity to the influence of this factor, which supports my point in Q1. At the same time, for the etth1 data set, as the number of experts increases, the effect becomes better . Why not continue to set more experts and explore the performance limits of this data set?

**Questions:**

See Weaknesses

---

### Official Review · Reviewer_d7sQ · 2024-11-02

**Soundness:** 2
**Presentation:** 2
**Contribution:** 2
**Rating:** 3
**Confidence:** 4

**Summary:**

The authors present a mix-of-expert linear model guided by a large language model (LLM). Their experiments demonstrate that LeMoLE achieves superior performance with a reduced computational footprint compared to existing LLM-based approaches.

**Strengths:**

- The proposed methodology is clear and easy to comprehend.

- The ablation study focusing on the frequency domain and the number of experts offers valuable insights.

**Weaknesses:**

- The novelty of the approach appears to be limited. The fundamental aspect of the model consists of a series of linear projections with varying input lengths. While the model retains a mixture of experts (MoE) architecture, it substitutes the aggregation component with a condition generated by an LLM.

- The utilization of text seems rather basic. For the dynamic aspect, it appears that only timestamps are considered, while for the static aspect, descriptions are provided for the channels.

Minor: Line 199, Equation (??)

Minor: Table 5, in caption the inference and training speed is in (s) but in the table it is marked as (ms).

**Questions:**

- It appears that LeMoLE performs better on the Traffic and Electricity datasets, which are significantly larger than the ETT dataset. Does this suggest that LeMoLE requires more data for effective training?

- The incorporation of multimodal information is promising; however, its effectiveness remains unclear. Specifically, using the timestamp alone as the dynamic input seems to provide similar information to the "time encode" employed in Autoformer and Fedformer, which translates timestamps into one-hot encodings. What advantages does using timestamp text as auxiliary input offer over this method?

- Additionally, the static information consists of channel descriptions, which should remain consistent across the training, validation, and test sets within the same dataset. This input may not contribute additional information and could limit generalization, serving primarily as a channel identifier.

- It is noted that LeMoLE exhibits a comparable number of parameters to MoLE at H=96, yet significantly exceeds it at H=720. Given that the MoE architecture is largely unchanged and the size of the GPT-2 encoder remains constant, could the authors clarify the source of the additional parameters?

---

### Official Review · Reviewer_hK8o · 2024-11-03

**Soundness:** 3
**Presentation:** 3
**Contribution:** 2
**Rating:** 3
**Confidence:** 4

**Summary:**

Linear models have proven effective in time series forecasting due to their capacity to capture and leverage the linear relationships inherent in many time series datasets. The challenge is to develop a powerful prediction model that retains the high efficiency of linear models. The authors develop an LLM-enhanced Mixture-of-Linear-Experts (MoLE) for time series forecasting. The authors mention that this is the first work on improving linear time series models based on mixture-of-expert learning and multimodal learning. Compared to several recent state-of-the-art prediction networks on long-term forecasting and few-shot tasks, this method demonstrates the effectiveness of this proposed LeMoLE in terms of accuracy and efficiency.

**Strengths:**

1. By integrating LLM with linear experts, especially multiple linear experts on time series forecasting tasks, LeMoLE achieve a new sota compared to existing LLM-based time series forecasting models;
2. This study is the first work that uses linear time series models based on mixture-of-expert learning and multimodal learning.

**Weaknesses:**

This article looks incomplete.

1. The novelty is not strong enough. This architecture roughly assembles LLMs into a linear layer, and there is no clear presentation of the used CNN structure. It would be better to theoretically discuss how and why CNN can overperform other architecture and to provide evidence on how a lightweight CNN can fuse the linear features.
2. Some typos and errors should be revised. For example, in Eq. 3, $b_{i}$ should be $b_{m}$. "Based on Equation (??)" should be revised.
3. The presentation sounds ambiguous in some sessions. For example, LeMoLE mentioned the use of varying window lengths, but in this article, there is no detail on the CNN when setting the range of $T$. Especially, when $T=96$ and the number of expert is $5$, the convolutional block should be $3 \times 3$ or $7 \times 7$? Or other settings?
4. A lack of visualization results. For example, comparisons between time-based and frequency-based LeMoLE outputs, or visualizations of how different window lengths affect the results.
5. Finally, the results are not consistent to the experiment settings. For example, the length of $T$ is not consistent to the $H$ of Table 2 and Table 3. Please double check if this is a typo.

**Questions:**

1. LeMoLE can process the global and local text data, can you please show some examples of how well LeMoLE can integrate the prompt text with time series data?
2. What is $H$ in Eq.1?
3. What is the architecture of CNN in Figure 2?
4. What is the limitation of this work?
5. What is the difference between LeMoLE-F and LeMoLE-T in processing the input features?

---

### Official Review · Reviewer_vc42 · 2024-11-04

**Soundness:** 2
**Presentation:** 3
**Contribution:** 2
**Rating:** 5
**Confidence:** 4

**Summary:**

This paper presents an LLM-enhanced mixture of linear experts framework designed for time-series forecasting with multiple lookback periods. The model effectively incorporates multimodal information by integrating both global and local textual data during the ensemble process of various linear experts. The proposed approach demonstrates both high efficiency and strong predictive capability in standard and few-shot forecasting scenarios.

**Strengths:**

1. This paper offers a new perspective for LLM-enhanced time-series forecasting models. Unlike existing methods that aim to align LLMs directly with time-series models, the proposed method integrates multimodal information to improve the ensemble of multiple linear experts.

2. Experiments demonstrate promising results for both standard and few-shot forecasting scenarios, meanwhile showing the model's efficiency compared with most LLM-enhanced forecasting models.

**Weaknesses:**

1. Why do time series descriptions and timestamps convey non-linearity? What is the benefit of embedding timestamps using LLM compared to the time embedding method in MoLE?

2. The reported values in Table 2 differ significantly from those in prior studies. It is important to include explanations about the experimental settings if the numbers were not directly adopted from existing literature.

3. Why are exchange and weather datasets excluded from the evaluation since they both demonstrate non-stationarity?

4. The framework lacks an automated approach for selecting the optimal number of experts for new evaluation datasets. Which column in Table 9 corresponds to Table 1? For a new dataset, does the process require manually testing the number of experts from 1 to 5 to determine the best results?

5. There is a typo in Line 199: "Based on Equation".

**Questions:**

1. What are the main differences compared with MoLE when both static and dynamic prompts are removed from LeMoLE? The results of "w/o both prompts" in Table 4 still look quite different than MoLE.

2. The caption for Table 4 states that the results are for a prediction length of 336. However, the results for the ETTh dataset in Table 4 appear to align with a prediction length of 96, not 336, in Table 2.

---

### Note · Authors · 2024-11-26

I have read and agree with the venue's withdrawal policy on behalf of myself and my co-authors.